



# Airborne Lidar Measurements of Atmospheric $CO_2$ Column Concentrations to Cloud Tops made during the 2017 ASCENDS/ABoVE Campaign

Jianping Mao[1,2], James B. Abshire[1,2], S. Randy Kawa[2], Xiaoli Sun[2], Haris Riris[2]

[1]University of Maryland, College Park, MD 20740, USA
[2]NASA Goddard Space Flight Center, 8800 Greenbelt Road, Greenbelt, MD 20771, USA

*Correspondence to*: Jianping.Mao@nasa.gov

**Abstract**. We measured the column-averaged atmospheric $CO_2$ mixing ratio ($XCO_2$) to a variety of cloud tops with an airborne pulsed multi-wavelength integrated path differential absorption (IPDA) lidar during NASA's 2017 ASCENDS/ABoVE field campaign. Measurements of height-resolved atmospheric backscatter profiles allow this lidar technique to estimate $XCO_2$ to cloud tops as well as to the ground with accurate knowledge of the photon path-length. We validated these measurements with those from an onboard in situ $CO_2$ sensor during spiral down maneuvers. These lidar measurements were 2-3 times better than those from previous airborne campaigns, due to our using a wavelength step-locked laser diode source and a high-efficiency detector for this campaign. Precisions of 0.6 parts per million (ppm) were achieved for 10-s average measurements to mid-level clouds and 0.9 ppm to low-level clouds at the top of the planetary boundary layer. This study demonstrates the lidar's capability to fill in $XCO_2$ measurement gaps in cloudy regions and to help resolve the vertical and horizontal distributions of atmospheric $CO_2$. Future airborne campaigns and spaceborne missions with this capability can be used to improve atmospheric transport modeling, flux estimation, and carbon data assimilation.

## 1. Introduction

Atmospheric carbon dioxide ($CO_2$) is a long-lived greenhouse gas that is widely transported. Globally distributed atmospheric $CO_2$ concentration measurements with high-precision, low-bias, and full seasonal sampling are essential to advance carbon cycle sciences and assess carbon-climate changes (Schimel et al., 2016). However, about two thirds of the Earth's surface is usually covered by clouds. High-quality retrievals of column-averaged atmospheric $CO_2$ mixing ratio ($XCO_2$) can only be attained from passive remote sensing measurements of $CO_2$ from space for clear-sky scenes without significant aerosol loading, where the path-length of the Earth's surface reflected sunlight is accurately known. Hence passive measurements of $XCO_2$ are significantly limited in spatial coverage and seasonal sampling, which may cause large uncertainty in regional and hemispheric carbon flux estimates (Chevallier et al. 2014; Reuter et al., 2014; Feng et al., 2009; 2016a, b). New observations to fill these gaps can be used to help improve carbon balance estimates (Palmer et al., 2019; Vekuri et al., 2023).



NASA Goddard Space Flight Center developed an airborne pulsed, integrated-path differential absorption (IPDA) lidar approach to measure $XCO_2$ as a candidate for NASA's planned Active Sensing of $CO_2$ Emissions over Nights, Days, and Seasons (ASCENDS) orbital mission (Abshire et al., 2010; Kawa et al., 2010; 2018). Concurrent measurements of height-resolved atmospheric

backscatter profiles allow this lidar technique to estimate $XCO_2$ and range to cloud tops in addition to those to the ground with precise knowledge of the photon path-length even in dense, broken, and sometimes multi-layered atmospheric clouds (Ramanathan et al., 2015; Mao et al., 2018; 2021a). This is a major advantage of this lidar approach over passive ones for measuring greenhouse gases when the elevation of the reflecting surface is uncertain (e.g., due to rough

terrain or tall trees) and when the atmosphere has significant scattering (Mao and Kawa, 2004; Aben et al., 2007).

The airborne version of our IPDA lidar has been flown on the NASA DC-8 aircraft five times since 2011 over a variety of sites in the U.S. and Canada to demonstrate instrument measurement capabilities and for regional science campaigns (Abshire et al., 2013; 2014; and 2018). We

previously demonstrated its capability to measure $XCO_2$ to cloud tops and the partial column $XCO_2$ between the ground and cloud tops by using a cloud slicing approach with data from the 2011, 2013, and 2014 airborne campaigns over the West and Midwest regions of the U.S. (Ramanathan et al., 2015; Mao et al., 2018). In 2014, we replaced the lidar's wavelength-swept seed laser source with a rapidly tunable step-locked seed laser (Numata et al., 2012). In 2016, we

replaced the PMT-based photon-counting receiver with a much more sensitive HgCdTe avalanche photodiode (APD)-based receiver (Sun et al., 2017). These updates substantially improved the lidar's dynamic range, stability, and signal-to-noise ratio, and reduced the measurement bias and increased precision (Abshire et al., 2018). This paper describes this lidar $XCO_2$ measurements made to cloud tops during the summer 2017 ASCENDS/ABoVE (Arctic

Boreal Vulnerability Experiment) campaign using the most recent instrument configuration (Mao et al., 2019 and 2021b). Most flights were based at Fairbanks, Alaska. The lidar's $XCO_2$ measurements are validated against those from onboard in situ sensors during spiral down maneuvers that were made nearby.

## 2.  Lidar Measurements of $XCO_2$

The airborne $CO_2$ Sounder lidar deployed in the 2017 airborne campaign used a tunable narrow line-width laser to measure $CO_2$ absorption at 30 wavelengths distributed across the vibration-rotation line of $CO_2$ centered at 1572.335 nm. The parameters of the airborne $CO_2$ lidar for the 2017 flights are the same as those for the 2016 flights and have been summarized in previous publications (Abshire et al., 2018; Sun et al., 2021). Briefly the laser emits 1 µs wide rectangular

pulses at a rate of 10 kHz. The laser scans across the $CO_2$ line with 30 wavelengths at a 300 Hz rate. The laser wavelengths were offset-locked to the center of this $CO_2$ absorption line by using a reference gas cell at a pressure of 40 hPa and a temperature of 296 K (Numata et al., 2011 and 2012). The laser wavelength step size varied from 250 MHz near the line center to 2.75 GHz on the wing, which allowed for well distributed samples across the line. The laser linewidth is



approximately 30 MHz or 0.001 cm$^{-1}$. The laser's spectral resolution is considerably higher than that of passive measurements, for example, GOSAT/GOSAT-2 (~0.2 cm$^{-1}$; Kuze et al., 2009), OCO-2/OCO-3 (~0.3 cm$^{-1}$; Crisp et al., 2004) and the ground-based Fourier Transform Spectrometers of the Total Carbon Column Observing Network (~0.02 cm$^{-1}$; Wunch et al.,

2011). The narrow laser linewidth allows the measured $CO_2$ line shape to be fully resolved, including the line width and the center wavelength (Ramanathan et al., 2013). The lidar's $XCO_2$ retrievals have sub-ppm sensitivity to $CO_2$ change in the measurement column and are independent of a priori $CO_2$ information, e.g., vertical distribution of $CO_2$ (Ramanathan et al., 2018).

The laser photons backscattered from the atmosphere and ground are collected by a 20-cm receiver telescope, pass through a narrow (~ 1 nm) band-pass filter, and then are focused onto the lidar's HgCdTe detector. The electrical bandwidth of the receiver is 8 MHz, and the receiver digitizer has a sampling period of 10 ns, allowing a vertical resolution of 1.5 m, providing accurate measurements of photon path-lengths. Previous campaigns showed range measurement

to better than 0.25 m to flat surfaces over a horizontal path from the laboratory and to better than 3 m to water surfaces on a near nadir path from the aircraft (Amediek et al., 2013). The range-resolved backscatter profiles are computed from off-line laser wavelengths after averaging data over 1 second and using a 15-m vertical layer thickness to improve the signal-to-noise ratio (SNR).

The lidar retrieval algorithm to estimate $XCO_2$ uses a weighted least-squares fit of the calculated $CO_2$ absorption line shape to the 30 wavelengths of the lidar measurement (Ramanathan et al., 2018; Sun et al., 2021). The fitting approach also allows simultaneously solving for Doppler frequency shift, surface reflectance at the off-line wavelengths, and the non-uniformity in the lidar's spectral response, which minimizes potential biases. The high spectral resolution and high

measurement sensitivity of this approach allows $XCO_2$ retrievals to be independent of a priori $CO_2$ information, e.g., vertical profile of $CO_2$, and inversion constraints.

In the retrieval forward calculations, the HITRAN 2008 spectroscopy database (Rothman et al., 2009) and the Line-By-Line Radiative Transfer Model (Clough et al., 1992; Clough and Iacono, 1995) V12.1 were used to calculate $CO_2$ optical depth for a prior with a vertically uniform $CO_2$

concentration of 400 ppm. The retrieval algorithm calculates the best-fit $XCO_2$ by comparing the calculated absorption line shapes to the lidar sampled line shapes and by uniformly scaling the calculated one to minimize the fit error. For this work where the retrievals were near a spiral down location, the retrievals used the DC-8 aircraft housekeeping data for the vertical profiles of atmospheric pressure and temperature, and water vapor profiles from an onboard engineering test

version of diode laser hygrometer (DLH; Diskin et al., 2002). When co-located radiosonde measurements were available within +/- 3 hours flight time, the radiosonde data of vertical profiles of atmosphere were used for forward calculations since radiosonde provides the best data about vertical structure of the atmosphere. The $XCO_2$ retrievals were primarily processed based on 1-s averaged lidar data, so since the DC-8 aircraft traveled horizontally at a speed of





240 m/s this resulted in a horizontal resolution of ~240-m along the ground track. When the DC-8 aircraft was in a spiral down maneuver, it descended at 7-8 m/s.

## 3. The 2017 ASCENDS/ABoVE Airborne Campaign

During July and August 2017, NASA conducted the ASCENDS/ABoVE airborne campaign
using the NASA DC-8 aircraft. The flights occurred between July 20 and August 8, 2017, over the ground tracks shown in Figure 1. In all, eight flights were conducted over the Central Valley of California, over the Midwest then to Fairbanks AK, over the Northwest Territories in Canada, and south and central Alaska (Mao et al., 2019 and 2021b), and then returning to California. This was the first time that airborne $XCO_2$ lidar measurements had been made over the arctic
region.

Compared to previous airborne campaigns, the 2017 airborne campaign was conducted in much more dynamic atmosphere conditions over the Northwest Territories of Canada and over Alaska and it overflew more clouds at multiple levels as well as fire smoke plumes. The $CO_2$ Sounder lidar continuously measured column absorption of $CO_2$ from the aircraft altitude to the ground
and to cloud tops, along with height-resolved backscatter profiles (Sun et al., 2021).

During the campaign, a total of forty-seven vertical spiral down maneuvers were conducted over a variety of atmospheres and surface types like desert, vegetation, permafrost, and the Arctic and Pacific Oceans. The purpose of these vertical spiral maneuvers was to compare the lidar $XCO_2$ retrievals with those computed from the onboard in situ $CO_2$ sensors.

The $XCO_2$ retrievals from the lidar measurements were validated against those computed from $CO_2$ vertical profiles measured in situ by the AVOCET sensor during the spiral down maneuvers. AVOCET has a stated precision of ±0.1 ppm (1-sigma) and accuracy of ±0.25 ppm (Halliday et al., 2019). The DC-8 aircraft housekeeping data provided temperature, pressure, geolocation, and positioning such as altitude and pitch/roll angles at flight altitude. The aircraft
radar altimeter also provided an independent range measurement to ground under all conditions since the radar measurement penetrates clouds and dense smoke plumes. Using the radar altimeter data with aircraft housekeeping data allows calculating the radar-measured surface elevation. This allows distinguishing the cloud tops from ground or ocean surface in the processing and analysis of the lidar measurements.

## 4. Case Studies

Three case studies with spiral down maneuvers nearby cloudy regions were selected and analyzed in this study. These are the flight segments over Grand Island, Nebraska on July 27[th], Inuvik, Northwest Territories of Canada on Aug. 3[rd], and over Bettles, Alaska on Aug. 6th.

### 4.1 Resolving Vertical Gradient of Atmospheric $CO_2$





We conducted a 9.4 hour long south-to-north flight on July 27, 2017, transiting from Palmdale, CA to Fairbanks, AK. We conducted spiral down maneuvers at four local airports during the flight. The first spiral down maneuver had a duration of about 20 minutes and was conducted over Grand Island, NE at 18:12 UTC or 1:12 PM local time from a flight altitude of 10 km to

near ground. The backscatter profile for this segment of the flights is shown in Figure 2 and a subsection of the $XCO_2$ in Figure 3. A very significant drawdown of $CO_2$ (~ 30 ppm) was observed near the surface at this site; the $CO_2$ mixing ratio at surface was as low as 376 ppm, while the average $CO_2$ mixing ratio in the free troposphere was 406 ppm (Figure 4). Some cirrocumulus clouds were near the aircraft altitude prior to the spiral maneuver, while during the

spiral down the sky was clear. During the flight out from Grand Island, the aircraft flew over altocumulus clouds for about 30 minutes (18:48-19:18 UTC or 1:48-2:18 PM local time). The cloud top heights of these mid-level clouds ranged from 5 to 7 km, as seen in Figure 2.

### 4.1.1 $XCO_2$ Measurements to the Ground

Figure 4 shows the comparison of lidar $XCO_2$ retrievals to the ground with those from the

AVOCET in situ sensor during this spiral down maneuver. The AVOCET instrument sampled the $CO_2$ mixing ratio outside the aircraft every second. The lidar $XCO_2$ retrievals were based on 1-s averaged lidar data. The in situ $XCO_2$ was computed from the integral of the AVOCET $CO_2$ vertical profile using the vertical averaging kernel of lidar's $XCO_2$ retrieval made from the same altitude. The comparisons were made for averages in every 1-km vertical layer of atmosphere

with more than 100 samples as DC-8 aircraft spiraled downward at about 7-8 m/s.

As the figure shows, the lidar's averaging kernel peaks in the planetary boundary layer, which means the lidar $XCO_2$ retrievals have the most weighting for $CO_2$ at the bottom atmospheric layers and which allows good sensitivity to surface fluxes. As shown in Figure 3, there was a significant difference between the lidar $XCO_2$ retrievals made to the ground and those to cloud

tops. Both in situ and lidar $XCO_2$ showed a strong vertical gradient, caused by a significant surface drawdown in this area that was covered with growing corn and soybeans crops. When DC-8 aircraft flew away from this area, the lidar $XCO_2$ increased steadily.

The retrievals with highest precision (lowest standard deviation from the least squares fit) were from flight altitude of 7-9 km, indicating the lidar's optimal operating altitude. The optimum is a

result of the combined effect of $CO_2$ differential absorption and number of returned laser photons. At higher flight altitudes there is more $CO_2$ absorption, but there are fewer returned laser photons. This causes a lower signal-to-noise ratio, and noisier $XCO_2$ retrievals that have larger standard deviations. At lower flight altitudes the reflected laser return is greater, but the photon path lengths are shorter, and the $CO_2$ absorption is much weaker, also causing the $XCO_2$

retrievals to have larger standard deviations. The 7-9 km altitude range is where there is the best balance between the line absorption and the number of received signal photons for this instrument.



Compared to the in situ $XCO_2$ from AVOCET, the lidar $XCO_2$ had an average bias of 0.1 ppm for flight altitudes above 5-km. These results are based on 1-s averaged lidar data which typically have a standard deviation of 1.2 ppm. When the lidar data averaging time is increased to 10 seconds, the standard deviation of lidar $XCO_2$ retrievals to the ground was 0.7 ppm. For 10

seconds flight time the length of the aircraft's ground track was typically 2.4 km. The longer averaging time improves the signal-to-noise ratio of the lidar data; however, it also increases the lidar range variation for non-flat surfaces, e. g., vegetation cover and cloud tops, which makes $XCO_2$ retrievals noisier (Mao et al., 2018). The overall benefit of longer data averaging time is to improve the precision of lidar $XCO_2$ retrievals, especially over flat surfaces like deserts and

oceans. Longer averaging times also benefit lidar XCO2 retrievals to cloud tops, as shown in the next section.

### 4.1.2 XCO₂ Measurements to Cloud Tops

During the flight out from the Grand Island spiral, the DC-8 flew for about 30 minutes over extended altocumulus clouds with cloud top heights between 5 to 7 km. As shown in Figure 3,

the difference between the lidar $XCO_2$ retrievals to the ground and those to cloud tops was significant, due to the surface drawdown in this area. As the DC-8 aircraft flew away from this area, the lidar $XCO_2$ increased steadily. Figure 5 shows the comparison of lidar $XCO_2$ retrievals to these mid-level cloud tops during the profile out with those from the in situ vertical profiles of $CO_2$ measured during spiral down segment for the flight altitudes above 7-km. The lidar range to

cloud tops was between 2 to 4 km. This was shorter than the optimum range for this lidar. From the flight altitudes 7-9 km, the lidar $XCO_2$ retrievals to cloud tops had an average difference of +0.8 ppm, compared to those measured by the in situ sensor during the spiral.

When DC-8 aircraft further flew away from Grand Island, this difference increased to +2.9 ppm. These larger differences are thought to result from significant horizontal differences in the

atmosphere (temperature, water vapor, and $CO_2$ profiles) between the region of the spiral down maneuver and that for the flight out segment. Note that the optical depth look-up-table used for retrievals was based on the vertical profiles of atmosphere measured during the spiral down. Retrievals of $XCO_2$ to cloud tops during the flight out were made using the same look-up-table for the spiral down. However, the actual conditions are expected to be somewhat different since

during summer the atmospheric conditions and $CO_2$ concentrations are expected to have significant gradients in this area.

The retrievals of $XCO_2$ to these cloud tops had a standard deviation of 3-4 ppm for 1-s averaged data, which was about 3 times larger than that for the retrievals to the ground. One reason for this is that the cloud reflectance at the lidar wavelength was typically about 5%, a factor of 4-5 times

lower than from vegetated surfaces (Mao et al., 2018). Calculations from the lidar data showed that median reflectance of these mid-level clouds at Grand Island was only 2.7% while median value of ground reflectance at Grand Island was 27%. Additionally, there was less $CO_2$ absorption in the shorter range and the effects together caused these lidar retrievals to be noisy.





In addition, the variability in elevation of cloud tops may degrade $XCO_2$ retrievals. The centroid cloud top altitudes were calculated from time-averaged lidar range measurements and used as photon path-length for retrievals to cloud tops. The variable range to these cloud tops also caused the $XCO_2$ retrievals to be noisier (Mao et al., 2018).

5    When the data were averaged over 10 seconds, the standard deviation of $XCO_2$ retrievals to these altocumulus cloud tops at Grand Island improved to 1.3 ppm. This measurement precision for partial column $XCO_2$ to cloud tops is at least two times better than those from the 2011, 2013, and 2014 airborne campaigns (Abshire et al., 2014; Mao et al., 2018). This improvement was caused by the utilization of the step-locked laser diode source and the high-sensitivity detector in 10    this campaign (Abshire et al., 2018).

## 4.2 Validation of Lidar $XCO_2$ Measurements to the Tops of Mid-level Cloud

After the flight from Palmdale, CA to Fairbanks, AK, we conducted two flights based out of Fairbanks. to the Northwest Territories of Canada (NWT). Both flights targeted a northern loop of the area, including the Arctic Ocean. Figure 6 shows the ground track, satellite image and 15    lidar backscatter profiles for the 2nd flight in NWT on Aug. 3 UTC time. This flight started late on Aug. 2 and went from Fairbanks to Inuvik, then east, then back north along the Arctic Ocean, back to Inuvik then back to Fairbanks. We again used spiral down maneuvers for comparing the lidar measurements of $XCO_2$ against those from the in situ $CO_2$ profiles above the airports at Inuvik, Kugluktuk, Cambridge Bay, Inuvik again, and then returned to Fairbanks.

20    As shown in Figure 6, the atmospheric conditions from Fairbanks to Inuvik varied from mostly cloudy to broken clouds at multiple levels on the return leg. These cloud layers provided opportunities for lidar cloud slicing (Ramanathan et al., 2015). There were several occurrences of thick cirrus clouds below DC-8 aircraft that attenuated the lidar signal and caused some data outages. The path over the Arctic Ocean was also very cloudy and the vertical structure of the 25    clouds was complex.

Also shown in Figure 6 are dense smoke plumes from wildfires in the south seen after the spiral down maneuvers at Inuvik and Cambridge Bay. The atmospheric scattering and attenuation caused by the smoke would significantly degrade or completely screen out any retrievals from passive spectrometers on satellites (Mao and Kawa, 2004; Aben et al., 2007; Butz et al., 2009; 30    Uchino et al., 2012; Guerlet et al., 2013). In contrast the lidar can accurately measure $CO_2$ enhancements from wildfires through dense smoke plumes, as demonstrated earlier for the large wildfires in the Canadian Rockies during this airborne campaign (Mao et al., 2021a). Measurements of height-resolved atmospheric backscatter profiles allow this lidar approach to accurately estimate $XCO_2$ and range to terrain and water surfaces even in the presence of 35    wildfire smoke.

During the first spiral down maneuver of the flight at Inuvik, NT, we overflew some altocumulus clouds with cloud tops around 4.5 km above ground (Figure 6). Figure 7 shows the comparison





of both $XCO_2$ retrievals to the ground and to these mid-level cloud tops against those from the in situ $CO_2$ profiles. The measurement local time was around 8 PM and evidence of a small surface sink was noticeable. The differences between the lidar $XCO_2$ retrievals to the ground and to the altocumulus cloud tops were -0.1 and +0.4 ppm, respectively, compared to those from the in situ $CO_2$ profile. The standard deviation of lidar's $XCO_2$ retrievals to the ground from flight altitude of 7-8 km was 1.3 ppm, while the standard deviation of $XCO_2$ retrievals to cloud tops from flight altitude of 8-9 km was 1.7 ppm. On average the lidar range to cloud tops was 4 km for this segment. The lidar measurements showed that the ground reflectance at Inuvik airport was ~ 30% at the lidar wavelength and that to the tops of the altocumulus clouds was 5.6%, more than twice that of clouds over Grand Island. This higher reflectance improved the precision of the lidar's $XCO_2$ measurements to these clouds.

These retrieval results were based on 1-s averaged lidar data. When the lidar data averaging time was increased to 10 seconds, the standard deviation for both retrievals to the ground and to cloud tops decreased to 0.6 ppm. The lidar's measurement precision to cloud tops indicates the benefit of measurement capability over persistent cloud cover – for example that which occurs over the west coasts of continents with marine layered clouds and over the Southern Ocean. These results show that averaging lidar measurements to cloud tops for a longer distance in these regions can fill these significant gaps with high-precision measurements.

Figure 8 shows the time series of lidar $XCO_2$ retrievals made to the ground and to the altocumulus cloud tops during this spiral down using 10-s data averaging. While the $XCO_2$ measurements to cloud tops were steady during this segment, the measurements to the ground exhibited lower values of $XCO_2$. This small 1.3 ppm difference between these two sets of measurements indicates slightly lower carbon below clouds likely caused by a small local $CO_2$ drawdown during summertime.

## 4.3 Validation of Lidar $XCO_2$ Measurements to Low-level Clouds

Measurements of $XCO_2$ to the ground and to the tops of nearby clouds at the top of the planetary boundary layer provides information to help separate the carbon processes at the Earth's surface from the carbon transport in the free troposphere (Mao et al., 2018; Shi et al., 2021). In earlier airborne campaigns we used broken cumulus clouds and demonstrated a lidar cloud slicing approach to estimate partial column $XCO_2$ in the planetary boundary layer (Ramanathan et al., 2015). However, the results from the earlier version of this airborne lidar had larger biases and large standard deviation even though the lidar data were aggregated over 10 or even 100 seconds (Abshire et al., 2014; Ramanathan et al., 2015; Mao et al., 2018).  The lidar used in the 2017 campaign had several hardware improvements that resulted in improved measurement performance.

After the two flights in the Northwest Territories of Canada we conducted two flights in the south and central Alaska. On August 6 the flight track went in a counterclockwise direction from



Fairbanks westward to Kotzebue, then almost due south, and on a diagonal path back toward Fairbanks (Figure 1). We again used spiral down maneuvers above the airports at Bettles, Kotzebue, Unalakleet, Platinum, McGrath, Fort Yukon, and Fairbanks to validate the lidar $XCO_2$ measurements.

The takeoff time of the August 6[th] flight was 7:45 AM local time and the spiral down at Bettles, AK started around 8:45 AM or at 16:54 UTC. As shown in Figure 9, the DC-8 aircraft flew over broken cumulus clouds for about 35 minutes prior to and during the Bettles spiral down. The heights of cumulus cloud tops ranged from 2 to 2.5 km above ground at the top of planetary boundary layer. As shown in Fig. 10 the in situ sensor showed the $CO_2$ concentration near the
surface was as high as 436 ppm in the morning, which was confined within the lowest 100-m layer. Above that layer the $CO_2$ concentration increased with altitude in the bottom 4-km and remained almost uniform in the upper layers. The $XCO_2$ values to the ground and to the tops of cumulus cloud for flight altitudes above 5-km were about the same.

Figure 10 shows the profile comparison with the in situ measurements. The $XCO_2$ retrievals
from lidar measurements to the ground and to the tops of cumulus cloud showed an average bias of +0.2 and -0.4 ppm, respectively, for flight altitudes above 5-km, compared to the in situ measurements. The standard deviations of $XCO_2$ measurements to the ground and to cloud tops were 1.5 ppm and 2.5 ppm, respectively, for 1-s average lidar data. For this case, the lidar reflectance of cumulus clouds was 6%, while the ground reflectance near the Bettles airport was
20 25%.

Figure 11 shows the time series of lidar measurements. It shows that the lidar $XCO_2$ retrievals to cloud tops were more scattered than those to the ground, which is mainly caused by the lower reflectance of clouds at the lidar measurement wavelength. Compared to the $XCO_2$ measurements to the mid-level altocumulus cloud tops, the $XCO_2$ measurements to the boundary
layer cumulus cloud tops were significantly noisier due to the puffy cumulus cloud tops and longer range from aircraft to cloud tops (Mao et al., 2018).

When the lidar data are averaged over 10 seconds, the standard deviation of $XCO_2$ measurements to the ground is 0.8 ppm and the standard deviation for $XCO_2$ to the cumulus cloud tops is reduced to 0.9 ppm. These lidar $XCO_2$ measurements to the tops of the low-level clouds from the
2017 airborne campaign are 2-3 times better than those from our previous airborne campaigns using the earlier version of the lidar (Mao et al., 2018).

## 5. Discussion and Summary

The 2017 ASCENDS/ABoVE airborne campaign was the first time that lidar measurements of $XCO_2$ had been extended to the Arctic region. The summertime Arctic atmosphere contained a
variety of cloud types whose tops were at different elevations. These conditions allowed the opportunity to perform lidar $XCO_2$ retrievals to cloud tops and to validate these measurements with those from the onboard in situ sensor during spiral down maneuvers.



The results showed the standard deviation of $XCO_2$ lidar retrievals to cloud tops for 1-s average lidar data from this campaign was equivalent to that for 10-s average data from previous campaigns in 2011, 2013, and 2014. The improvement in data precision for this campaign was caused by the utilization of a step-locked laser diode source and the higher-sensitivity lidar

detector.

When the data averaging time was increased to 10 seconds, the standard deviations of the lidar retrievals improved to 0.6 ppm for the mid-level clouds and 0.9 ppm for the low-level clouds at the top of the planetary boundary layer. The $XCO_2$ measurements to cloud tops were typically 2-3 times noisier than those to the ground due to the lower reflectance of clouds at the 1572 nm

lidar measurement wavelength. During the 2017 airborne campaign most flight altitudes were below 10 km and so the lidar ranges to cloud tops were relatively short. There were many occurrences of cirrus clouds during the flights, however the ranges from the aircraft to the tops of these cirrus clouds were short resulting in weak $CO_2$ absorption and poor retrievals. For future space-based lidar measurements, the higher orbit altitude and longer atmospheric path length to

cirrus clouds should allow useful $XCO_2$ measurements to cirrus clouds as well.

These results indicate the significant benefit of the lidar's measurements to cloud tops, particularly those made over persistent cloud covers, e.g., the Inter Tropical Convergence Zone, west coasts of continents with marine layered clouds, Southern Ocean with low-level clouds, and the Arctic. These are important areas with active carbon cycling but where measurements from

passive satellite-based spectrometers are sparse or unavailable.

This study demonstrated that this lidar's $XCO_2$ measurements to cloud tops along with those to the ground can be used to help resolve vertical and horizontal gradients of $CO_2$. This lidar capability can be used to fill significant measurement gaps left by passive spectrometer missions and to help resolve the vertical distribution of atmospheric $CO_2$. Future airborne campaigns and

spaceborne missions with this lidar measurement capability, like NASA's planned ASCENDS mission (Kawa et al., 2018). would improve carbon data assimilation, atmospheric transport modeling, flux estimation, and advance carbon cycle science.

**Data availability**. The lidar $XCO_2$ retrievals for the clear sky from the 2017 airborne campaign is available from the NASA Airborne Science Data for Atmospheric Composition website,

https://www-air.larc.nasa.gov/cgi-bin/ArcView/ascends.2017#ABSHIRE.JAMES/ (NASA Langley Research Center, 2020). The lidar $XCO_2$ retrievals to the cloud tops used in this work are available from the primary authors.

**Author contributions.** Dr. Mao led the manuscript writing, the airborne campaign data processing and analysis. Dr. Abshire was the principal investigator of the $CO_2$ Sounder lidar

development and led the 2017 ASCENDS/ABoVE airborne campaign. Dr. Kawa contributed to the 2017 airborne campaign, the campaign data processing and analysis, and the manuscript editing. Dr. Sun contributed to the $CO_2$ Sounder lidar development, the lidar data analysis, and



the manuscript editing. Dr. Riris contributed to the $CO_2$ Sounder lidar development, the 2017 airborne campaign, and the campaign data processing and analysis.

**Competing interests:** The authors declare that they have no conflict of interest.

**Acknowledgement:** This work was supported by the NASA ABoVE project, the NASA ASCENDS mission pre-formulation activity, and NASA's Airborne Science Program. We appreciate the work of Graham Allan, Kenji Numata, and Jeffrey Chen from NASA Goddard for supporting the $CO_2$ Sounder lidar as well as Joshua P. DiGangi, Glenn Diskin, and Yonghoon Choi from NASA Langley Research Center for participating in the airborne campaign and for providing the in situ $CO_2$ and $H_2O$ data. We gratefully acknowledge the work of the DC-8 aircraft team at NASA's Armstrong Flight Center for helping to plan and conduct the flight campaign. We thank Paul T. Kolbeck from University of Maryland at College Park for the lidar backscatter data processing. We acknowledge the use of imagery from the NASA Worldview application (https://worldview.earthdata.nasa.gov/), part of the NASA Earth Observing System Data and Information System (EOSDIS).

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



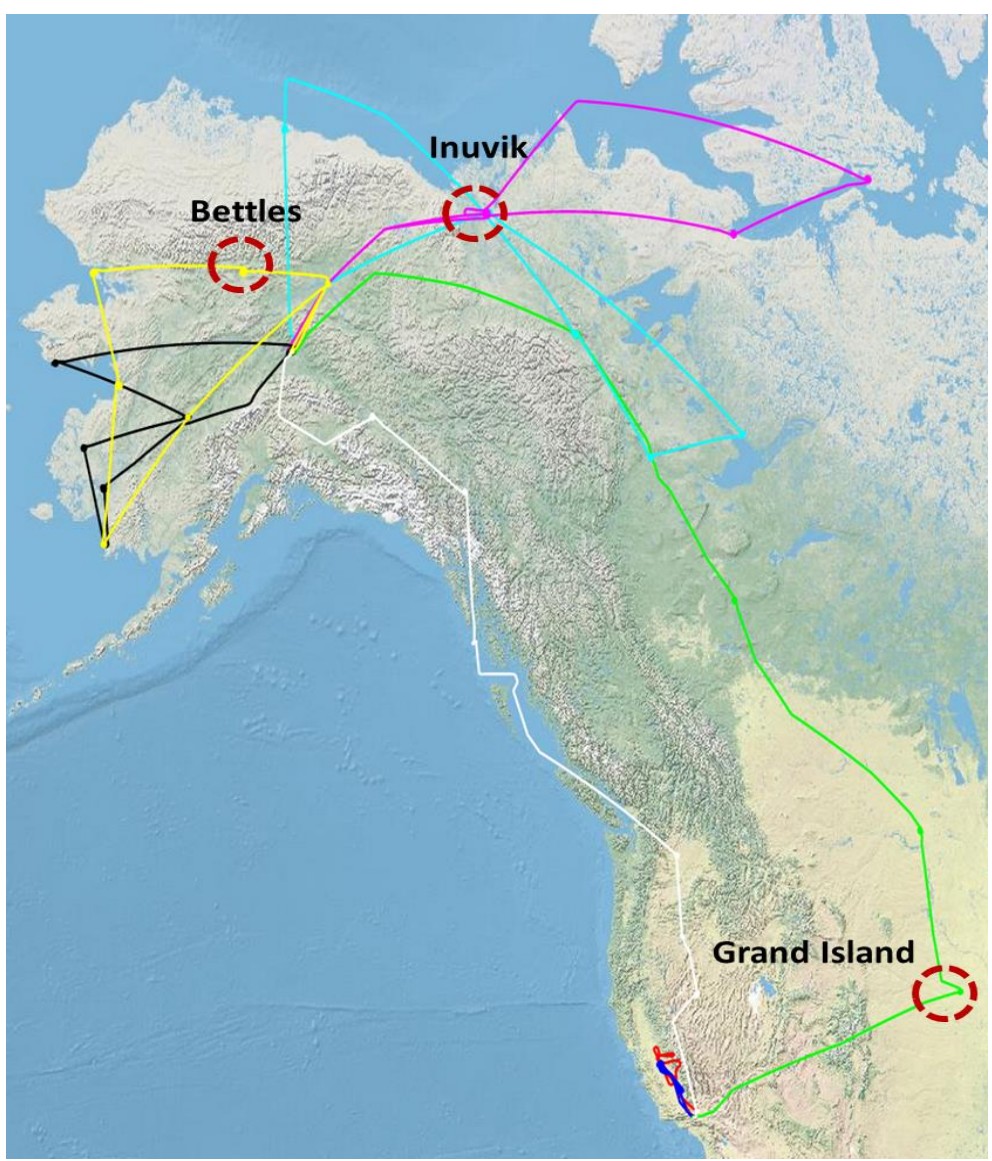

Figure 1. Map of flight tracks for the 2017 ASCENDS/ABoVE airborne science campaign with NASA DC-8 aircraft (© Google Maps 2019). The colors of the track indicate a total of eight flights from July 20th to August 8th. The three spiral maneuvers are marked in red circles for the three cases described in this study over Grand Island, Nebraska on July 27[th], Inuvik, Northwest Territories of Canada on Aug. 3[rd], and Bettles, Alaska on Aug. 6th.

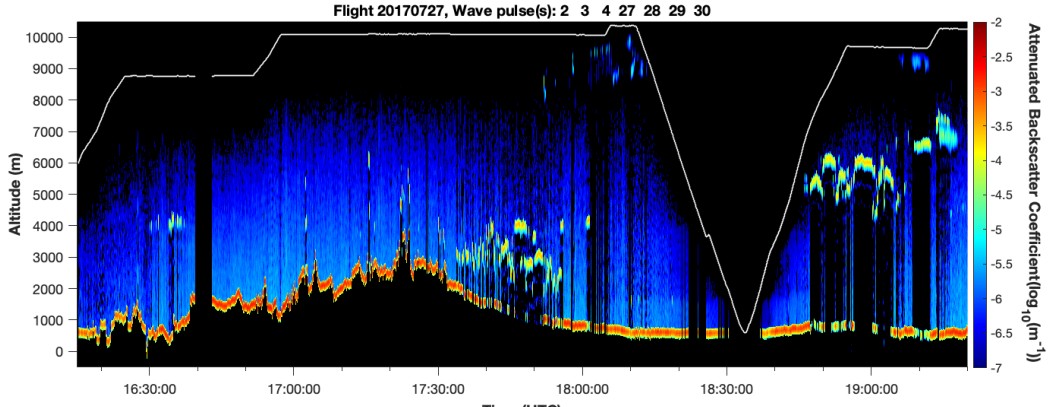

Figure 2. Time series of the range-corrected attenuated backscatter profiles measured for the flight over the Rocky Mountains and spiral down over Grand Island, NE on July 27, 2017. The measurements have 1-s time resolution and a vertical resolution of 15-m. The GPS flight altitudes are marked in a white line and ground elevation is shown in the red and yellow band. The lidar returns are averaged for offline wavelengths or wave pulses # 2, 3, 4, 27, 28, 29, and 30 on the wings of the $CO_2$ absorption line. The strong returns from the ground and clouds are colored in yellow and red, while the lidar returns from aerosols and cirrus clouds are weaker and plotted in light blue.



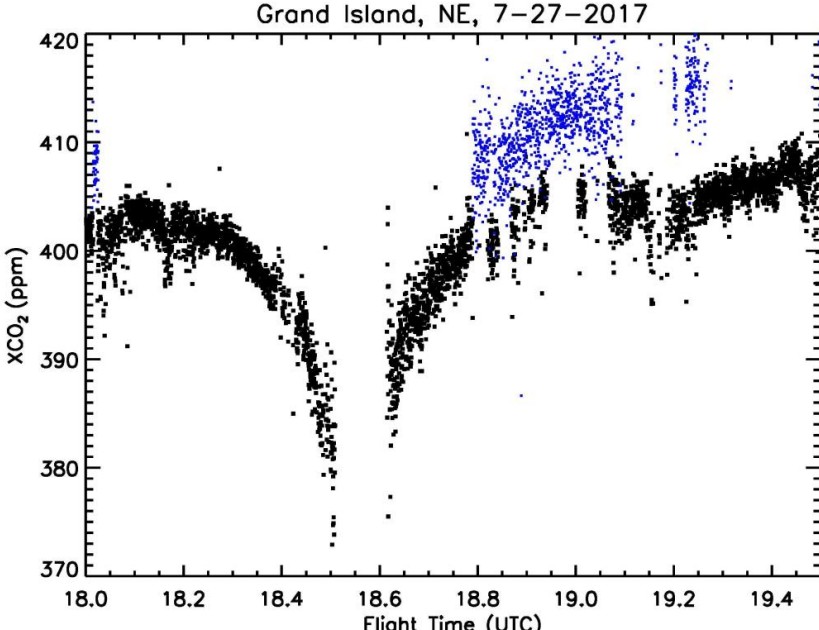

Figure 3. The time series of $XCO_2$ retrievals from lidar measurements made surrounding the spiral at Grand Island, NE on July 27, 2017, using 1-s averaging. The black dots are the retrievals from the lidar measurements to ground and blue dots are those made to the tops of altocumulus clouds.



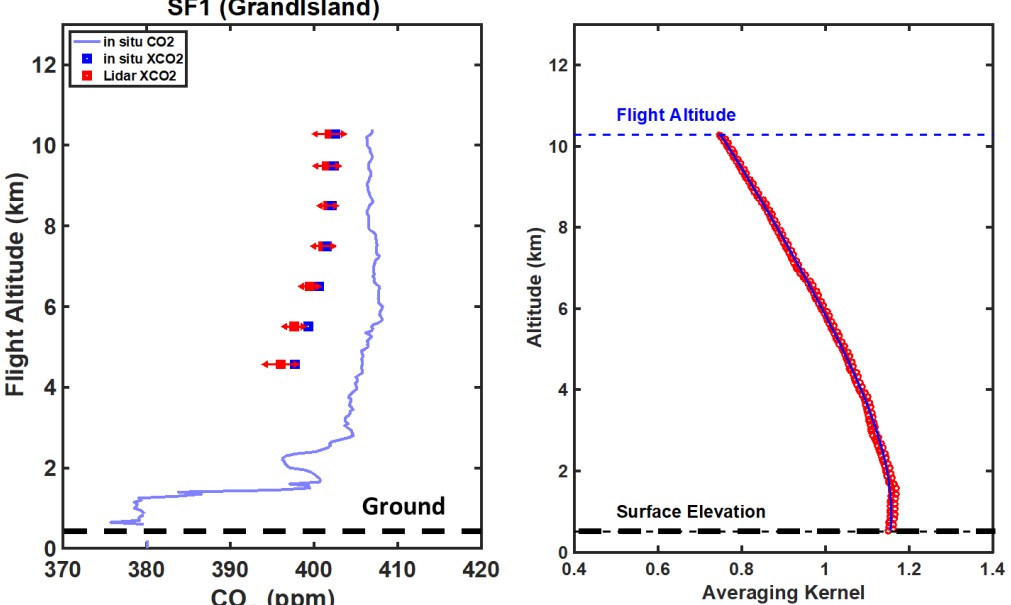

Figure 4. Comparison of cloud-free lidar $XCO_2$ retrievals to the ground with those from in situ measurements made during the spiral maneuver at Grand Island, NE on July 27, 2017. These are computed as a function of flight altitude for averages in every 1-km vertical layer of atmosphere. The $XCO_2$ computed from in situ values are marked in blue squares and the values of the lidar's $XCO_2$ retrievals are marked in red squares. The red error bars for the lidar's $XCO_2$ retrievals are ±1 standard deviation. One of the vertical averaging kernels for the lidar's $XCO_2$ retrievals for this profile segment is shown at the right. The ground is marked in thick black dashed line at the bottom and the flight altitude is marked in a blue dotted line at the top.



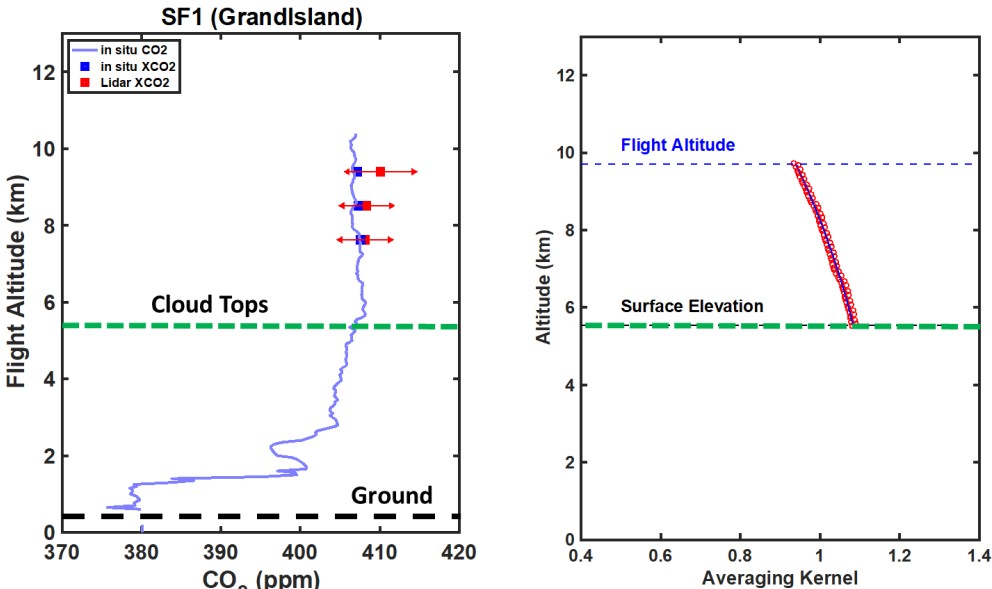

Figure 5. Same as Fig. 4 but for the comparison of XCO2 retrievals from lidar to cloud tops with those from in situ measurements during the flight ascent from Grand Island, NE. The ground is marked in a thick black dashed line at the bottom and the average cloud top height is marked in a thick green dashed line.



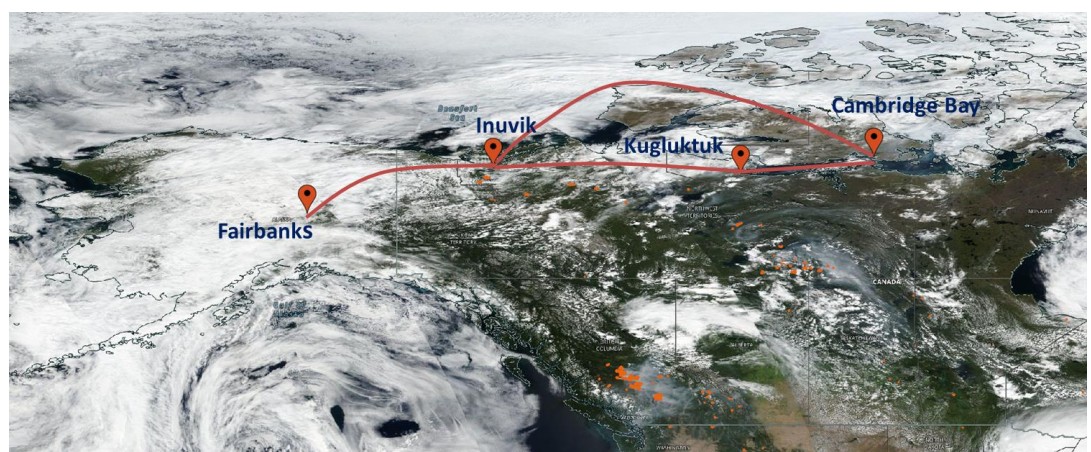

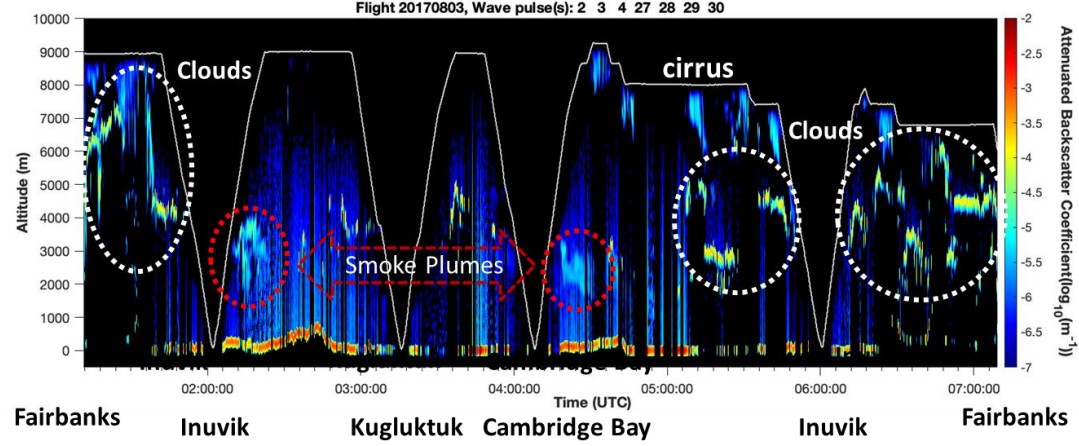

Figure 6. True color image from Aqua/MODIS (top; NASA Worldview) and time series of the
lidar's attenuated backscatter profiles (bottom) for the flight over the Northwest Territories,
Canada on Aug 3, 2017. Clouds are in white, and wildfires are marked in red dots in the MODIS
color image. Clouds, including cirrus, and wildfire smoke plumes are circled and labeled in the
lidar profiles. The flight ground track is marked in a red line in the top image and the aircraft
GPS flight altitude is marked in a white line in the bottom plot. The locations of spiral maneuver
are labeled. The lidar range-corrected attenuated backscatter profiles were sampled at a vertical
resolution of 15-m and averaged over 1-s. Several occurrences of cirrus clouds are clearly seen
as light blue regions just below the aircraft's altitude.

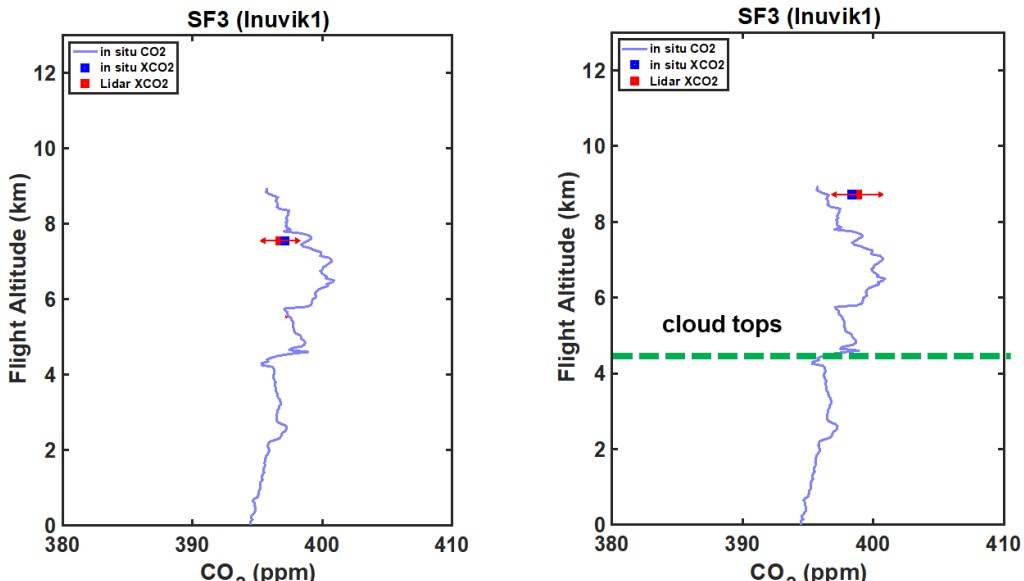

Figure 7. Measurements during the first spiral down maneuver at Inuvik, NT on Aug. 3, 2017.
Comparison of $XCO_2$ retrievals from lidar measurements made to the ground (left) and the $XCO_2$
retrievals from lidar measurements to cloud tops (right). The $CO_2$ profile measured with the in
situ sensor is plotted as the blue line. The lidar measurements used 1-s averaging. The in situ
$XCO_2$ values are marked in blue squares and the lidar's $XCO_2$ retrieval values are marked in red
squares. $XCO_2$ values were binned into the top 1-km vertical layer of atmosphere. The red error
bars for the lidar $XCO_2$ retrievals are ±1 standard deviation. In the plot on the right the average
cloud top height is marked as a green dashed line.



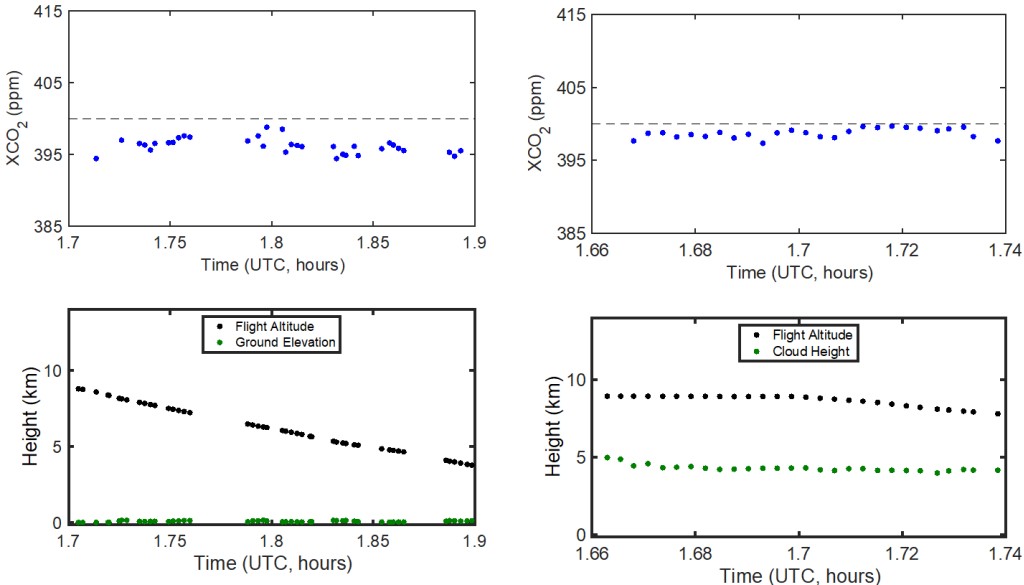

Figure 8. Retrievals from lidar measurements during the spiral down maneuver at Inuvik, NT on
Aug. 3, 2017, using 10-s averaging. Upper Left: the time series of the $XCO_2$ retrievals made to
the ground (blue dots). Lower left : DC-8 aircraft altitudes and ground elevation for the same
segment. Upper and lower right: Same as for the left but for the lidar $XCO_2$ retrievals made to
cloud tops (blue dots). In the lower figures the flight altitudes are plotted in black dots and the
elevation of the ground and cloud tops used for the lidar measurements are plotted in green dots.

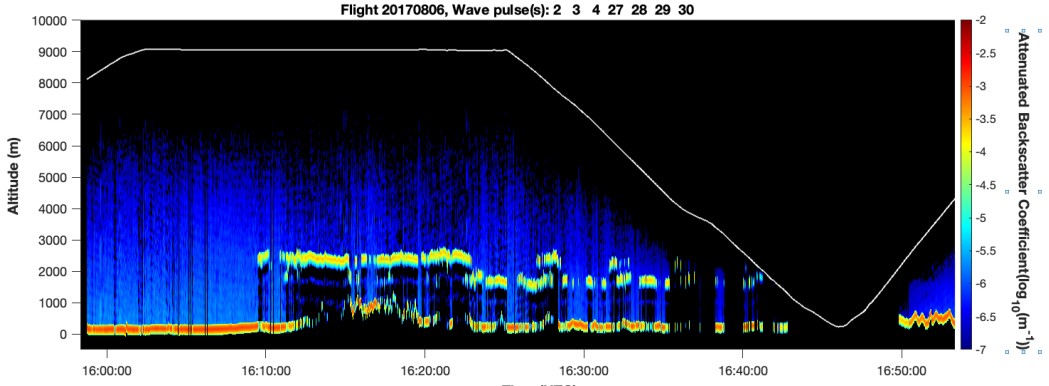

Figure 9. The time series of the lidar's range-corrected attenuated backscatter profiles measured for the flight segment over Bettles, AK on Aug. 6, 2017. The lidar measurements are averaged for 1-s and have 15-m vertical resolution. The lidar returns from cloud tops are comparable to those from ground as indicated by their red and yellow color scale of the attenuated backscatter coefficients. The aircraft's flight altitude is marked as a white line.

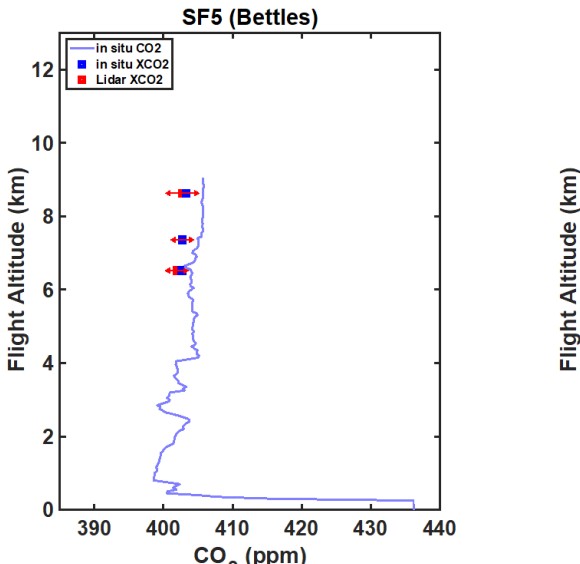
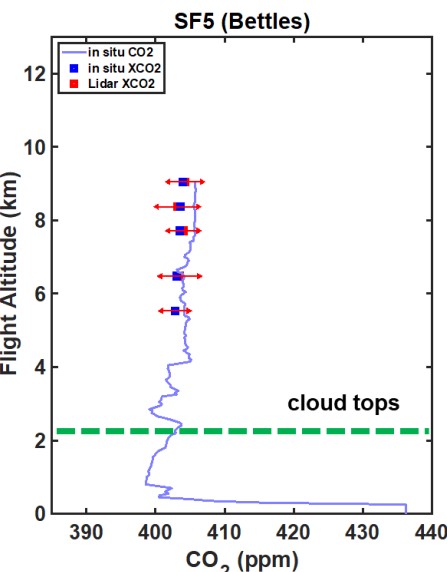

Figure 10. $XCO_2$ retrievals from lidar measurements near the spiral down over Bettles, AK on Aug. 6, 2017. The $XCO_2$ retrieved from 1-s averaged lidar measurements to the ground is on the left and the lidar $XCO_2$ retrievals to cloud tops in on the right. The $CO_2$ profile measured from the in situ sensor during the spiral down maneuver is plotted as the blue line. The $XCO_2$ values computed from the in situ measurements are marked as blue squares and the lidar's $XCO_2$ retrieval values are marked in red squares. $XCO_2$ values were binned into the top 1-km vertical layer of the atmosphere. The red error bars for the lidar retrievals are ±1 standard deviation. In the plot on the right the average cloud top height is marked as a green dashed line.



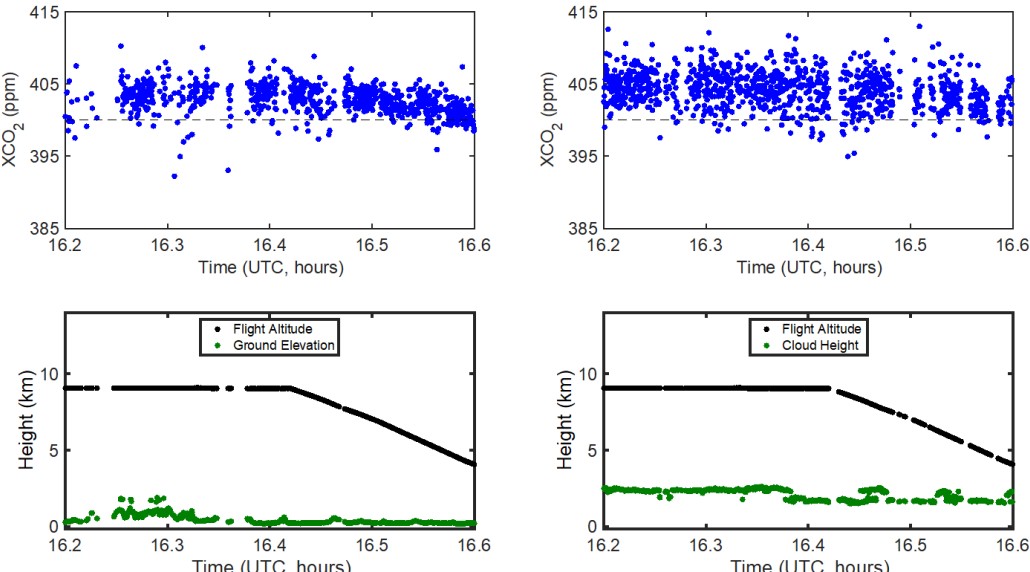

Figure 11. Left: The time series of the $XCO_2$ retrievals measured during the spiral down maneuver over Bettles, AK on Aug. 6, 2017. Upper Left: The blue dots are $XCO_2$ measured to the ground for 1-s averaged lidar data. Lower Left: the DC-8 aircraft altitudes and ground elevation. Upper and lower right: same as the lefthand plots but for the lidar $XCO_2$ retrievals to cloud tops (blue dots). In the lower figures the flight altitudes are plotted in black dots, and the ground elevations and cloud top heights are plotted in green dots.