# Peer review of "Airborne Lidar Measurements of Atmospheric CO2 Column Concentrations to Cloud Tops made during the 2017 ASCENDS/ABoVE Campaign"

_Atmospheric Measurement Techniques, 2023_

## Author Comment (AC1)

We appreciate your thorough review and helpful comments.

We have accepted all your editorial comments and made corrections in the revision of the manuscript. The official name of this instrument is called 'CO$_2$ Sounder lidar'.

The following are our one-to-one responses to your comments/questions.

General Comment from reviewer: This is an important work that should be published in this journal, after the points noted below are addressed. The main point of concern is that the definition of the column-integrated quantity of interest -- "XCO$_2$" -- appears to be different here than from how it is normally used. When the term "XCO$_2$" is used in the satellite CO$_2$ measurement community, or by those researchers who use Earth-based Fourier transform spectrometers (e.g., as part of the TCCON and COCCON networks) who validate such measurements, it is defined as the mass-weighted dry-air mole fraction: moles of CO$_2$ per moles of dry air. The contributions to the column integral come equally by mass from all layers of the atmosphere. The sensitivity of the instrument to the different layers (as defined by the averaging kernel vector) might vary, but contributions from an a priori estimate of the CO$_2$ profile are used to make up the difference: if the instrument has low sensitivity to CO$_2$ in the upper part of the column (values of the averaging kernel vector close to zero), then most of the information on CO$_2$ for that part of the column integral is taken from the prior; conversely, if the opposite is true near the surface (values of the averaging kernel vector close to one), then little information needs to be taken from the prior. Here, however, the column integral addressed does not appear to have equal weight (by mass) from each layer of the atmosphere, when computing the average. Instead, it appears that the weight of each layer is taken directly from the averaging kernel vector, normalized so that the total across all layers sums up to one, for whatever portion of the total column is being measured. No contribution from an a priori CO$_2$ profile is used to make the column-integral mass-weighted. There is nothing wrong with using this flavor of vertical weighting to present the results here. However, if the authors want to assign the name "XCO$_2$" to it, they must at minimum note how their definition differs from the one commonly used. In my opinion, it would be better to use some other symbol to denote the vertical weighting used here. [If I am wrong and the authors are actually using a flat mass-weighted vertical integral, then they must do a better job of describing their method in the manuscript, so that the reader can understand what they are doing.] The second point that the authors should address is how they remove the effect of water vapor from their column integral. Since XCO$_2$ is usually defined using the mass of DRY air (water vapor removed), then the contribution of water vapor to the mass of each layer, when doing the vertical weighting, must be removed. The authors should address this in their method discussion. If they do not remove the effects of water vapor, then they should note this when they define the vertical integral that they are using (i.e., what they call "XCO$_2$" at the moment). This point is repeated below, to some extent, in the detailed comments.

Response: The approach and meaning of our lidar measured XCO$_2$ has not different from others, ours is clearly mass-weighted, or optical-depth-weighted in the retrieval (Ramanathan et al., 2018; Sun et al., 2021). Our averaging kernel (AK) is the same as that defined in Borsdorff et al. (2014) for profile scaling-based retrieval. We used a uniform XCO$_2$ profile for all layers in our retrieval. The AK is proportional to the partial derivative of the retrieved column XCO$_2$ scale

factor with respective to the actual $XCO_2$ in each layer. The AK was also used to integrate the in situ $CO_2$ measurements to obtain the in situ $XCO_2$ up to a given height for comparison with the lidar measurements. To update the manuscript, we added the following note in Section 2 (p. 3):

'Note that the averaging kernel in our retrieval is based on that defined in Borsdorff et al. (2014) for profile scaling-based retrieval, giving a measure of the sensitivity of the column scale factor to the $XCO_2$ in each layer. We used the averaging kernel for a uniform a priori $CO_2$ profile to compute the in situ $XCO_2$ with the in situ vertical profile of $CO_2$ to validate the lidar $XCO_2$ retrievals during spiral down maneuvers.'

Speaking of atmospheric water vapor, we did take this into account in the forward calculations and in the retrieval. We computed both $CO_2$ and $H_2O$ absorption to match measured line shape for the best $XCO_2$ with regards to dry air. We added the following sentences in Section 2 (p. 4),

'There is a weak isotopic water vapor (HDO) line centered at 1572.253 nm on the shoulder of the 1572.335 nm $CO_2$ line. Depending on atmospheric water vapor content, this can distort the $CO_2$ line shape and could significantly impact the value of the $XCO_2$ retrieval. Therefore, the real-time water vapor absorption was calculated and added to $CO_2$ absorption for the best absorption line shape fitting in the retrieval.'

Detailed comments from reviewer:

page 2, Section 2:  The authors need to do a better job defining the vertical weighting that goes into "$XCO_2$": if it is indeed different from the usual definition (see comments above), the authors need to be clear about this; they may want to consider using a different name to describe their quantity, to avoid confusion.  In particular, the authors appear to use a vertical weighting proportional to the averaging kernel vector, instead of the flat mass-weighting usually used for $XCO_2$.

Response: please see the previous response.

page 3, Line 18: It was not clear to me how "using a 15-m vertical layer thickness" relates to improving the SNR.  Maybe add some more details?

Response: The raw lidar signal is digitized at 10 ns (1.5 m) intervals by the ADC to enable meter-level calculations of the lidar range to the surface. However, the vertical resolution of the computed atmospheric backscatter profiles is much wider due to the 1 μs (150 m) laser pulse width. To reduce the data volume, the backscatter profile data is re-sampled with 100 ns (15 m) bin width. We reworded the sentence to 'To reduce the data volume, the backscatter profile data is re-sampled with 100 ns (15 m) bin width. The range-resolved backscatter profiles are computed from off-line laser wavelengths after averaging data over 1 s to improve the signal-to-noise ratio (SNR).'

p3 L47: "of atmosphere": What do you mean by this, exactly? Atmospheric temperature, pressure, and water vapor? Some subsets of these? Or additional variables, as well? It would be clearer to spell this out more precisely.

Response: We think you meant p3 L37, not L47. You are right that 'vertical profiles of atmosphere' used in the forward calculations include temperature, pressure, and water vapor. We changed this sentence to 'the radiosonde data of vertical profiles of atmospheric temperature, pressure, and water vapor were used for forward calculations,'

p5 -- the definition of "$XCO_2$" and the use of the averaging kernel (AK) is not clear. It appears that the AK defines the vertical weighting that is used in the average, and $XCO_2$ is a non-flat-with-pressure vertical average. The assumed a priori $CO_2$ profile is not used in the calculation of the vertical average. All this is fine, but it needs to be explicitly described, and the difference between the definition of $XCO_2$ here and elsewhere needs to be clarified. If $XCO_2$ is indeed its usual flat-with-pressure definition, then we need to know the a priori $CO_2$ profile used. Also, how is the effect of water vapor removed, if $XCO_2$ is versus dry air?

Response: please see the previous response about our lidar $XCO_2$ and AK.

p7 L1-4: These lines could be reworded to make the point clearer. Maybe something like: "In addition, variability in the elevation of the cloud tops may degrade the accuracy of the $XCO_2$ retrievals. The cloud top altitude used to calculate the photon path-length in the retrieval is taken as the centroid of the cloud top altitudes calculated from time-averaged lidar range measurements. These range measurements are not taken at the same time as the measurements across the center of the line and are also averaged across a longer time (1 second); the difference in range that results causes an additional error in the $XCO_2$ retrievals that causes greater noise (Mao et al., 2018)." I am not sure if I have captured the key point -- if not, please reword to make the idea clearer.

Response: We appreciate your in depth understanding. Following your recommendation, we reworded these lines to "In addition, the variability in the elevation of the cloud tops may degrade the accuracy of the $XCO_2$ retrievals. The cloud top altitude used to calculate the photon path-length in the retrieval is taken as the centroid of the cloud top altitudes calculated from time-averaged lidar range measurements. These range measurements are not taken at the same time as the measurements across the $CO_2$ absorption line and are also averaged across a longer time (1 second or 10 seconds); the difference in range results in an additional error in the $XCO_2$ retrievals (Mao et al., 2018)."

p8 L22-24: "This small 1.3 ppm difference between these two sets of measurements indicates slightly lower carbon below clouds likely caused by a small local $CO_2$ drawdown during summertime." $CO_2$ at these Arctic latitudes is mainly dominated by the signal of terrestrial

carbon and release from further south being transported up north.  I.e., it is not a local signal. For example, at the Barrow site, the large seasonal variability there is not due to a strong local terrestrial signal, but rather to the strong seasonal terrestrial at lower latitudes in the northern hemisphere being transported to the north by winds and mixing.  Please reword to reflect this.

Response:  Yes, we agree with your statement. We shortened the statement to 'This small 1.3 ppm difference between these two sets of measurements indicates slightly lower carbon below clouds' without exploring the causes.

Section 4.3:  If these low-cloud measurements can be used to isolate $CO_2$ in the layer under the clouds, why have you not tried to use your measurements to calculate this?  I imagine this is because the large positive $CO_2$ values measured by the in situ sensor just close to the surface make assigning an average value for the thicker layer under the clouds less meaningful, but it would be useful for you to note what your reasons were.

Response: In earlier airborne campaigns we used broken cumulus clouds and demonstrated a lidar cloud slicing approach to estimate partial column $XCO_2$ in the planetary boundary layer (Ramanathan et al., 2015). That was a case with a large vertical gradient of $CO_2$. For this case from the 2017 airborne campaign, the vertical gradient of XCO2 was small, as shown in Fig. 10. The high $CO_2$ confine in the lowest 100 m doesn't significantly contribute to $XCO_2$ below clouds. Therefore, it was not meaningful to use these $XCO_2$ retrievals with a few tenth ppm biases to resolve a small vertical gradient below and above clouds in this case.

p10 L13-15: "For future space-based lidar measurements, the higher orbit altitude and longer atmospheric path length to cirrus clouds should allow useful $XCO_2$ measurements to cirrus clouds as well."  It is not the orbit altitude or the path length that matters here, but rather the mass of the atmosphere (since $CO_2$ is well-mixed) between the top of the atmosphere and the 10 km flight referenced here.  For the Arctic, where the tropopause is low, that atmospheric mass is small, so that there might not be much of an improvement between what is seen from the satellite and these flights.  Therefore, I am not sure that the assertion that useful measurements above cirrus should be possible from satellites is valid, at least not in the Arctic, where there is not much atmospheric mass above 10 km.

Response:  You make a very good point. Air mass is a factor for $CO_2$ absorption; pressure broadening is another factor for wavelength dependence of absorption. At the tropopause, the pressure is only 20% of sea-level pressure, or 200 mb.  According to our line-by-line radiative transfer calculations, the $CO_2$ absorption of the 20% air-mass for 2-way pathlength is still very substantial on this measurement line, particularly at the line center. The weighting function for the line center wavelength peaks at the lower stratosphere, while the weighting functions for wavelengths at line wings peak at the surface. Satellite $XCO_2$ retrievals to cirrus using the absorption features near the line center could be useful for retrieving $XCO_2$ above tropopause and resolving vertical structure of $CO_2$. We demonstrated one case from our 2013 airborne

campaign, showing a clear measured $CO_2$ absorption line shape for cirrus at 10.5 km from an aircraft altitude of 12.1 km (see Fig. 13 of Mao et al., 2018, which is also attached here).

[Figure]

Figure 13. A $CO_2$ absorption line shape measured on March 7, 2013, to cirrus cloud tops at 10.5 km altitude (left panel). The lidar measurements are the blue circles and the fitted line shape is the solid black line. AVOCET in situ vertical profile of $CO_2$ concentration is plotted at the right panel. The aircraft flight altitude was 12.1 km and the lidar range to cirrus cloud tops was 1.6 km.

References: put the journal names in italic script, and the volume numbers in bold?

Response: Although this convention was used by some journals, it looks like it is not used by this journal.

Whole document: if the journal style permits, put Latin phrases like "in situ" and "a priori" in italic script.

Response: we italicized 'a priori' but make 'in situ' normal as this phrase is commonly used in this field.

Whole document: remove the dash sign between number and unit?

Response: We did this following your recommendation in the updated manuscript.

---

## Author Comment (AC2)

We appreciate your review and comments.

We have corrected the two language issues on p. 2 and 7 in the revised version of this manuscript, as you recommended.

Your remark on p. 6 was also a language issue; the word 'noisier' in this sentence was not a good choice. We reworded it to 'have larger standard deviations.' For the longer data averaging times under constant observing conditions the measurement signal-to-noise ratio improves. However, at the same time the average measurement footprint is longer and for these flights over clouds the lumpy cloud tops cause larger variations in the lidar range. This causes the standard deviations of the individual $XCO_2$ retrievals to increase. The net result of these two competing effects for measurements to cloud tops with longer averaging times was to reduce the standard deviation of the $XCO_2$ retrievals.